# ProCoSA: Probabilistic Concept Learning with Spatial Alignment

## Abstract

Concepts are human-interpretable semantic units that enable intervenable intermediate representations in vision models. However, acquiring concept annotations is expensive and typically incomplete, limiting scalable interpretability. We propose **ProCoSA**, a probabilistic framework that treats missing concepts as latent variables and jointly infers concept posteriors and task predictions under partial supervision. To enhance spatial coherence and reduce pseudo-label bias, **ProCoSA** introduces a spatial alignment prior that encourages concept activations to align with salient image regions, yielding more calibrated concept probabilities for downstream reasoning. The framework integrates seamlessly into existing concept-to-task pipelines without relying on any specific bottleneck architecture. Experiments on four benchmark datasets under low concept supervision show that **ProCoSA** consistently matches or surpasses state-of-the-art methods on both concept and task performance under identical evaluation protocols. The code will be released upon acceptance.

## 1 Introduction

Deep neural networks have achieved remarkable success across a wide range of domains (LeCun et al., 2015; Senior et al., 2020), yet their internal mechanisms often remain opaque (Samek et al., 2021) and may rely on unintended or undesired features (Achtibat et al., 2023). This lack of transparency poses challenges for deployment in high-risk and regulation-sensitive scenarios (Rudin, 2019; Haibe-Kains et al., 2020). As a result, explainable artificial intelligence (XAI) has gained increasing attention as a means to better understand model behavior and decision rationale (Došilović et al., 2018; Černevičienė & Kabašinskas, 2024). While local XAI methods such as saliency maps highlight "where" the model attends, they often fail to convey "what" semantic evidence the model has recognized (Kindermans et al., 2017). Concept-based explanations address this limitation by introducing human-interpretable concepts as intermediate representations that clarify which semantic features influence model predictions (Bau et al., 2017).

Despite their advantages, concept-based representations typically require inserting a set of human-defined concepts at a bottleneck, and real-world applications often suffer from incomplete or missing concept annotations (Koh et al., 2020). This sparsity undermines accurate modeling of the concept space and limits scalability in practice. To reduce annotation costs, prior work explores unsupervised or semi-supervised approaches, such as prompting LLMs to propose concepts or using heuristic pseudo-label propagation (e.g., $k$NN) (Yang et al., 2023; Hu et al., 2024). However, these pipelines usually bypass explicit modeling of the concept prediction function and offer no principled way to quantify uncertainty over missing concepts, making them fragile under sparse supervision and limiting both generalization and interpretability. Moreover, reliance on LLMs introduces additional concerns regarding stability, reliability, and transparency.

To address these limitations, we propose **ProCoSA**, a probabilistic framework for concept learning with spatial alignment. ProCoSA treats missing concept labels as latent variables and jointly infers concept probabilities and task predictions under partial supervision. To enhance spatial consistency and mitigate pseudo-label bias, we introduce a spatial alignment prior that encourages concept activations to focus on salient input regions. In contrast to heuristic pseudo-labeling, ProCoSA performs principled posterior inference over missing concepts via an Expectation–Maximization (EM) pro-

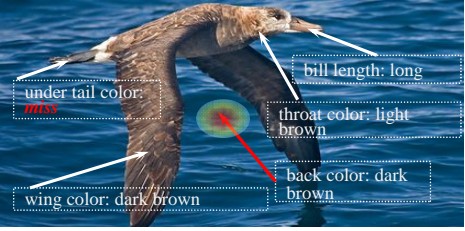

Figure 1: **Motivation illustration.** Left: Complete attribute annotations correctly aligned with corresponding visual regions. Right: Missing attribute supervision (e.g., "under tail color: miss") leads to spatial misalignment, where the model incorrectly links the semantic concept to irrelevant regions (highlighted in red), resulting in biased concept learning.

cedure, yielding more robust and interpretable learning under incomplete supervision. Figure 1 illustrates the core motivation behind our approach. Our contributions are as follows:

- We propose ProCoSA, a probabilistic framework for concept learning under partial annotations that treats missing concepts as latent variables and jointly learns concept and task predictions.
- We introduce a spatial alignment prior that guides concept representations toward salient regions, improving spatial consistency and reducing pseudo-labeling bias.
- ProCoSA yields calibrated concept inference and can be seamlessly integrated into existing concept-to-task pipelines without relying on any specific bottleneck architecture.

We evaluate ProCoSA on four representative datasets under a unified evaluation protocol. Across all settings, ProCoSA matches or surpasses prior methods in both concept and task performance, with further improvements reflected in enhanced concept-level interpretability metrics, particularly when concept supervision is scarce.

## 2 RELATED WORK

**Concept-Based Model Interpretability.** Human-understandable concepts provide a consistent semantic reference and a structured intermediate representation for interpreting neural networks. Network Dissection quantifies unit-level interpretability by testing alignment between individual channels and human-defined concepts using pixel-level semantic masks and IoU scores (Bau et al., 2017), offering spatial localization. Testing with Concept Activation Vectors (TCAV) measures a model's global sensitivity to user-defined concepts by learning concept activation vectors in feature space and computing directional derivatives along them (Kim et al., 2018). Both methods are *post hoc*; they do not support concept-level intervention or handle missing concept labels. In addition, TCAV depends on analyst-curated concept sets, assumes local linear separability, and lacks uncertainty-aware reasoning. In contrast, CBMs make concepts an explicit intermediate representation and predict task labels from the predicted concepts (Koh et al., 2020), thereby enabling concept-level intervention. However, CBMs typically assume fully annotated concept labels during training, which is costly and often unrealistic in practice, limiting their applicability when concept annotations are missing.

**CBMs with Incomplete Concept Supervision.** Recent CBM variants reduce manual concept supervision by constructing concept banks with LLMs and CLIP-based vision–language alignment. Res-CBM augments a base concept bank with optimizable residual vectors and incrementally discovers new concepts, improving accuracy while remaining a *post hoc* method; however, it increases pipeline complexity, depends on CLIP, and requires additional curation of the candidate bank (Shang et al., 2024). Label-free CBM converts a pretrained network into a CBM by generating concepts with LLMs, aligning them to CLIP text embeddings, and training a sparse classifier on the induced concept activations; it scales and preserves accuracy but inherits the same external dependence (Oikarinen et al., 2023). LaBo generates sentential concepts with a language model, selects a discriminative and diverse bottleneck via a submodular objective, and aligns concepts to images with CLIP; it reduces manual supervision yet remains *post hoc* and externally dependent (Yang et al., 2023). Despite these advances, heavy reliance on external resources makes concept sets prompt- and

domain-sensitive, and text–image alignment can be unstable (Zhang et al., 2024). To avoid external generators, SSCBM assigns $k$NN pseudo-concept labels and aligns similarity-based pseudo-labels at the concept level, and it jointly trains on labeled and unlabeled data. This improves concept accuracy and saliency alignment under partial supervision. However, SSCBM relies on heuristic $k$NN propagation, which bypasses explicit modeling of the concept predictor and is sensitive to feature-space noise, and it lacks any uncertainty-aware treatment of missing concepts. As a result, pseudo-label errors may propagate and degrade performance (Hu et al., 2024).

In contrast, we adopt a probabilistic approach that treats missing concepts as latent variables and jointly infers concept posteriors and task predictions through EM algorithm. The E-step leverages a spatial alignment prior to produce uncertainty-calibrated concept estimates, while the M-step updates model parameters to improve task performance. This yields explicit concept predictors and enhances generalization under partial supervision.

## 3 METHOD

**Overview.** We begin by formalizing concept learning under partial supervision as a latent-variable model and deriving the associated training objective (Sec. 3.1). We then explain how missing concepts are inferred within an EM loop using a mean-field E-step (Sec. 3.2). Next, we incorporate a spatial alignment prior computed from cosine similarities between concept embeddings and spatial features, together with two lightweight regularizers: (i) an alignment-score calibration loss and (ii) a spatial-consistency entropy penalty (Sec. 3.3). Finally, we summarize the overall loss and optimization schedule that jointly train the concept head and the label predictor using both observed and inferred concept labels (Sec. 3.4). The complete training pipeline is illustrated in Fig. 2.

### 3.1 PROBLEM FORMULATION

Human-understandable intermediate representations, such as semantic concepts, have been introduced to improve interpretability and enable intervention in high-stakes applications. Instead of directly mapping inputs to task labels, this paradigm first predicts a set of interpretable concepts and then predicts the final label from these concepts. However, acquiring fully annotated concept labels is costly and often infeasible at scale. To address this challenge, we formulate concept learning under partial supervision as a probabilistic latent-variable problem, in which missing concept labels are treated as latent variables and inferred jointly with the task.

In this setting, an $L$-way classification task consists of a dataset $\mathcal{D} = \{(x_i, y_i, \tilde{\mathbf{C}}_i)\}_{i=1}^N$ with $N$ samples, where $x_i \in \mathcal{X} \subset \mathbb{R}^d$ is an input sample, $y_i \in \mathcal{Y} = \{1, \ldots, L\}$ is the ground-truth task label, and $\tilde{\mathbf{C}}_i \in \{0, 1, -1\}^K$ is a partially labeled concept vector, with $-1$ denoting a missing entry. For convenience, we introduce an observation mask $\mathbf{m}_i \in \{0, 1\}^K$ induced by $\tilde{\mathbf{C}}_i$, where $m_{ik} = \mathbb{I}[\tilde{C}_{ik} \in \{0, 1\}]$. Let $\mathbf{C}_i \in \{0, 1\}^K$ denote the underlying complete concept vector. When $m_{ik} = 0$, we treat $C_{ik}$ as a latent variable and marginalize over its possible values during training. The observation model relating $(\mathbf{C}_i, \mathbf{m}_i)$ to $\tilde{\mathbf{C}}_i$ is

$$\tilde{C}_{ik} = \begin{cases} C_{ik}, & \text{if } m_{ik} = 1, \\ -1, & \text{if } m_{ik} = 0, \end{cases} \tag{1}$$

so that $\tilde{\mathbf{C}}_i$ coincides with $\mathbf{C}_i$ on observed dimensions and uses $-1$ to indicate missing concepts. We also write $\mathbf{C}_i = (\mathbf{C}_i^{\text{obs}}, \mathbf{C}_i^{\text{mis}})$, where $\mathbf{C}_i^{\text{obs}} = \{C_{ik} \mid m_{ik} = 1\}$ and $\mathbf{C}_i^{\text{mis}} = \{C_{ik} \mid m_{ik} = 0\}$.

We model concept learning under partial supervision via the following latent data-generating process:

1. draw an input from the data distribution: $x \sim p(x)$;

2. draw a full concept vector from the concept head: $\mathbf{C} \sim p_{\theta_c}(\mathbf{C} \mid x) = \prod_{k=1}^K \text{Bernoulli}(C_k \mid f_{c,k}(x; \theta_c))$;

3. draw a label from the conditional distribution given the concept vector: $y \sim p_{\theta_y}(y \mid \mathbf{C}) = \text{Categorical}(y \mid f_y(\mathbf{C}; \theta_y))$.

Figure 2: **Training pipeline of ProCoSA with latent concept inference.** The top (labeled) branch encodes images into concept activations, passes them through the bottleneck, and predicts labels; training uses both task loss and concept loss on observed concepts. The bottom (unlabeled) branch performs latent concept inference: (1) a feature extractor produces spatial features; (2) cosine similarity between learned concept embeddings and spatial features yields a spatial alignment prior over image locations; (3) the E-step combines this prior with concept activations to estimate posteriors of missing concepts; and (4) the M-step updates the concept encoder and label predictor using both observed and inferred concept labels. The loop arrow denotes one EM cycle. Color legend: yellow = concept extraction, purple = label prediction, green = spatial alignment, blue = EM inference/update.

where $f_c(\cdot; \theta_c) : \mathcal{X} \to [0,1]^K$ is the concept predictor that outputs per-concept probabilities, and $f_y(\cdot; \theta_y) : \{0,1\}^K \to \Delta_{L-1} = \{\mathbf{v} \in [0,1]^L : \mathbf{v}^\top \mathbf{1} = 1\}$ is the task predictor that maps a concept vector to class probabilities. The product form assumes conditional independence across concepts given $x$. Here, $\theta_c$ and $\theta_y$ are trainable parameters of the concept and task predictors, respectively.

**Objective.** We learn $(\theta_c, \theta_y)$ by maximizing the marginal log-likelihood of the observed data:

$$\max_{\theta_c, \theta_y} \sum_{i=1}^{N} \log p_{\theta_c, \theta_y}(y_i, \mathbf{C}_i^{\mathrm{obs}} \mid x_i) = \max_{\theta_c, \theta_y} \sum_{i=1}^{N} \log \sum_{\mathbf{C}_i^{\mathrm{mis}}} p_{\theta_y}(y_i \mid \mathbf{C}_i) \, p_{\theta_c}(\mathbf{C}_i \mid x_i), \quad (2)$$

where the inner summation is taken over all completions of $\mathbf{C}_i$ that agree with the observed entries $\mathbf{C}_i^{\mathrm{obs}}$ (i.e., $C_{ik} = \tilde{C}_{ik}$ whenever $m_{ik} = 1$). This defines a latent-variable model with $\mathbf{C}_i^{\mathrm{mis}}$ as the latent variables. We therefore employ an EM schedule to maximize equation 2: within each mini-batch, we perform an E-step followed by one parameter update on $(\theta_c, \theta_y)$; see Sec. 3.2 for details.

## 3.2 Handling Missing Concept Annotations

Directly maximizing equation 2 is difficult because it requires marginalizing over the latent concepts $\mathbf{C}_i^{\mathrm{mis}}$. We therefore resort to the EM algorithm and maximize an evidence lower bound on equation 2. Let $q_i(\mathbf{C}_i^{\mathrm{mis}})$ denote a mean-field variational posterior (MFVI) for sample $i$, supported only on completions consistent with the observations. The EM Q-function for sample $i$ is

$$Q_i = \mathbb{E}_{q_i(\mathbf{C}_i^{\mathrm{mis}})} \Big[ \log p_{\theta_y}(y_i \mid \mathbf{C}_i^{\mathrm{obs}}, \mathbf{C}_i^{\mathrm{mis}}) + \log p_{\theta_c}(\mathbf{C}_i^{\mathrm{obs}}, \mathbf{C}_i^{\mathrm{mis}} \mid x_i) \Big]. \quad (3)$$

**Inferring Missing Concepts (E-step).** We use mean-field variational inference to approximate the intractable posterior $p(\mathbf{C}_i^{\mathrm{mis}} \mid x_i, y_i, \mathbf{C}_i^{\mathrm{obs}})$. These posteriors are anchored by concept-head predictions and, when available, by a spatial alignment prior computed from the same backbone's spatial features (see Fig. 2 and Sec. 3.3).

$$q_i(\mathbf{C}_i^{\mathrm{mis}}) = \prod_{k \in \mathcal{U}_i} q_{ik}(C_{ik}; \phi_{ik}), \qquad q_{ik}(C_{ik} = 1; \phi_{ik}) = \phi_{ik}, \quad (4)$$

where $\mathcal{U}_i = \{k \mid m_{ik} = 0\}$ and each $q_{ik}$ is a Bernoulli distribution with mean parameter $\phi_{ik} \in [0,1]$. The variational posterior $q_i$ is obtained by maximizing the evidence lower bound:

$$q_i^\star = \arg\max_{q_i} \mathbb{E}_{q_i} \big[ \log p_{\theta_y}(y_i \mid \mathbf{C}_i) + \log p_{\theta_c}(\mathbf{C}_i \mid x_i) \big] + \mathcal{H}(q_i), \quad (5)$$

where $\mathcal{H}(q_i)$ denotes the (Shannon) entropy. Equivalently, this can be written as the following minimization involving the Kullback–Leibler divergence:

$$q_i^\star = \arg\min_{q_i} \mathrm{KL}\big[q_i(\mathbf{C}_i^{\mathrm{mis}}) \,\big\|\, p_{\theta_c}(\mathbf{C}_i^{\mathrm{mis}} \mid x_i)\big] - \mathbb{E}_{q_i}\big[\log p_{\theta_y}\big(y_i \mid \mathbf{C}_i^{\mathrm{obs}}, \mathbf{C}_i^{\mathrm{mis}}\big)\big]. \quad (6)$$

Under the mean-field parameterization, the coordinate-wise optimum admits a logistic fixed-point update for each missing concept $k \in \mathcal{U}_i$:

$$\mathrm{logit}(\phi_{ik}) = \mathrm{logit}\big(p_{\theta_c}(C_{ik} = 1 \mid x_i)\big) + \psi_{ik}^{\mathrm{cons}} + \lambda_{\mathrm{align}}\, w_{ik}\, a_{ik}, \qquad \mathrm{logit}(p) \triangleq \log \frac{p}{1-p}, \quad (7)$$

where $\psi_{ik}^{\mathrm{cons}}$ is an optional concept-consistency prior, $a_{ik}$ is the alignment logit defined in Sec. 3.3, and $w_{ik} \in \{0, 1\}$ is a confidence/top-$\kappa$ gate (within the missing set $\mathcal{U}_i$) that activates the spatial prior only on missing entries. Observed concepts are clamped: if $m_{ik} = 1$, then $q_{ik}$ degenerates to a delta at $C_{ik} = \tilde{C}_{ik}$. In practice, we run $T_\mathrm{E}{=}5$ fixed-point iterations of equation 7 per E-step and apply mild label smoothing to clamped entries to avoid numerical instabilities when evaluating log-likelihood terms. A more detailed variational interpretation and theoretical analysis of our training procedure are provided in Appendix A.

**Updating Model Parameters (M-step).** Given the posteriors $q_i$, we maximize the completed objective with respect to $(\theta_c, \theta_y)$:

$$\begin{aligned}
\theta_c^{(t+1)}, \theta_y^{(t+1)} &= \arg\max_{\theta_c, \theta_y} \sum_{i=1}^{N} Q_i(\theta_c, \theta_y; q_i) \\
&= \arg\max_{\theta_c, \theta_y} \sum_{i=1}^{N} \Big\{ \mathbb{E}_{q_i}\big[\log p_{\theta_y}(y_i \mid \mathbf{C}_i)\big] \\
&\qquad + \mathbb{E}_{q_i}\big[\log p_{\theta_c}(\mathbf{C}_i^{\mathrm{mis}} \mid x_i)\big] + \log p_{\theta_c}(\mathbf{C}_i^{\mathrm{obs}} \mid x_i) \Big\}.
\end{aligned} \quad (8)$$

With the factorized Bernoulli concept head and a categorical task head, this decomposes into: (i) training $p_{\theta_c}$ using *soft* targets $\phi_{ik}$ for $k \in \mathcal{U}_i$ and *hard* labels $\tilde{C}_{ik}$ for $m_{ik} = 1$; and (ii) training $p_{\theta_y}$ with $\mathbf{C}_i$ replaced by its posterior mean $\mathbb{E}_{q_i}[\mathbf{C}_i]$ (or Monte Carlo samples), using cross-entropy on $y_i$. During training, we alternate $T_\mathrm{E}{=}5$ fixed-point E-updates with one parameter update; training proceeds for 100 epochs with early stopping. Section 3.3 augments equation 6 with a spatial alignment prior to regularize posterior inference under partial concept supervision.

## 3.3 Spatial Alignment Prior for Concept Inference

Under partial concept supervision, inferred posteriors can become biased and spatially inconsistent. To mitigate this, we introduce a *spatial alignment prior* within the iterative inference loop shown in Fig. 2. The key intuition is that a concept should be grounded in salient image regions; thus, spatial evidence extracted from the image can guide the variational posterior toward semantically plausible concept values when labels are missing.

We first describe how the spatial evidence is computed. Given $x_i$, the feature extractor $\Omega(\cdot)$ outputs a spatial feature map $\mathbf{V}_i \in \mathbb{R}^{H \times W \times m}$. The concept encoder produces concept activations $\hat{\mathbf{C}}_i = f_c(x_i; \theta_c)$, and the embedding backbone provides a bank of learnable, $\ell_2$-normalized concept embeddings $\{\hat{\mathbf{c}}_k\}_{k=1}^{K}$. For each concept $k$, we compute a per-location heatmap by cosine similarity between $\hat{\mathbf{c}}_k$ and the local descriptors $\mathbf{V}_{i,p,q}$:

$$H_{i,k}(p,q) = \frac{\hat{\mathbf{c}}_k^\top \mathbf{V}_{i,p,q}}{\|\hat{\mathbf{c}}_k\| \, \|\mathbf{V}_{i,p,q}\|}, \quad p = 1, \ldots, H, \; q = 1, \ldots, W. \quad (9)$$

Next we aggregate the heatmap into a single alignment score in a way consistent with our implementation: we use *softmax pooling with temperature* $\tau_\mathrm{a} > 0$. Let:

$$\alpha_{i,k}(p,q) = \frac{\exp\big(H_{i,k}(p,q)/\tau_\mathrm{a}\big)}{\sum_{u=1}^{H} \sum_{v=1}^{W} \exp\big(H_{i,k}(u,v)/\tau_\mathrm{a}\big)}, \qquad a_{ik} = \sum_{p=1}^{H} \sum_{q=1}^{W} \alpha_{i,k}(p,q) \frac{H_{i,k}(p,q)}{\tau_\mathrm{a}}, \quad (10)$$

where $a_{ik}$ is the *alignment logit* (its probability is $\pi_{ik} = \sigma(a_{ik})$). To avoid injecting unreliable priors, we activate the alignment only where it is needed and confident: the binary gate $w_{ik} \in \{0, 1\}$ is set to one if and only if the concept label is missing ($m_{ik} = 0$), the alignment probability $\sigma(a_{ik})$ exceeds a threshold $\tau$, and the concept is among the top-$\kappa$ missing concepts of that sample (within $\mathcal{U}_i$) according to $\sigma(a_{ij})$; otherwise $w_{ik} = 0$. Formally, the rule is:

$$w_{ik} = \mathbb{I}[m_{ik} = 0] \cdot \mathbb{I}[\sigma(a_{ik}) \geq \tau] \cdot \mathbb{I}[k \in \text{Top-}\kappa(\{\sigma(a_{ij})\}_{j \in \mathcal{U}_i})]. \tag{11}$$

*Hyperparameters.* All hyperparameters follow SSCBM and are kept fixed across runs (including the Top-$\kappa$ size $\kappa$ and threshold $\tau$).

We then inject the spatial evidence into the variational objective for missing concepts: specifically, we regularize the mean-field factors $q_{ik}$ toward a Bernoulli prior with mean $\pi_{ik}$ by a KL term, which gives the following E-step objective:

$$\max_{q_i} \; \mathbb{E}_{q_i}\big[\log p_{\theta_y}(y_i \mid \mathbf{C}_i) + \log p_{\theta_c}(\mathbf{C}_i \mid x_i)\big] + \mathcal{H}(q_i) - \lambda_{\text{align}} \sum_{k \in \mathcal{U}_i} w_{ik} \, \text{KL}\big(q_{ik} \,\|\, \text{Bernoulli}(\pi_{ik})\big),$$
$$\tag{12}$$

where $\lambda_{\text{align}} \geq 0$ controls the prior strength and $\mathcal{U}_i = \{k : m_{ik} = 0\}$ collects the missing concepts. Optimizing equation 12 under the mean-field family in equation 4 yields the fixed-point update already stated in equation 7: the posterior mean $\phi_{ik}$ is obtained from the concept-head logit, plus an additive alignment bias $\lambda_{\text{align}} w_{ik} a_{ik}$, optionally plus the concept-consistency bias $\psi_{ik}^{\text{cons}}$ introduced in Sec. 3.2. Observed concepts remain clamped to their labels. In practice we run a few fixed-point iterations per E-step (e.g., $T_{\text{E}}{=}5$) and apply mild label smoothing at clamped entries to keep log-likelihood terms numerically stable.

**Alignment supervision.** Since $a_{ik}$ directly contributes to the spatial alignment prior, its calibration critically affects posterior inference under sparse supervision. To improve its quality, we introduce a lightweight cross-supervision objective that does not interfere with the variational update. For observed entries ($m_{ik} = 1$), we supervise the sigmoid-normalized alignment score $\hat{p}_{ik} = \sigma(a_{ik})$ using the ground-truth concept label $C_{ik}^{\text{obs}}$. For missing entries ($m_{ik} = 0$), we supervise it using the soft label $\phi_{ik}$ inferred from the posterior distribution.

$$\mathcal{L}_{\text{align}} = \beta_{\text{align}} \, \mathbb{E}_i \sum_k \Big[ m_{ik} \, \ell_{\text{CE}}(\hat{p}_{ik}, C_{ik}^{\text{obs}}) + (1 - m_{ik}) \, \ell_{\text{CE}}(\hat{p}_{ik}, \phi_{ik}) \Big], \tag{13}$$

where $\ell_{\text{CE}}$ is binary cross-entropy and $\beta_{\text{align}} \geq 0$ is a time-ramped weight to avoid overly strong early regularization.

**Spatial consistency regularizer** $\mathcal{R}_{\text{spat}}$**.** If the alignment heatmap is overly diffuse, the prior $a_{ik}$ becomes less discriminative and injects spatial noise. To encourage concentration over salient regions, we penalize the entropy of the softmax-normalized heatmap $\alpha_{i,k} = \text{softmax}(H_{i,k}/\tau_{\text{a}})$ on "active" concepts (e.g., $p_{\theta_c}(C_{ik} = 1 \mid x_i) > \frac{1}{2}$):

$$\mathcal{R}_{\text{spat}} = \beta_{\text{s}} \, \mathbb{E}_i \sum_{k \in \mathcal{K}_i} \Big( -\sum_{p,q} \alpha_{i,k}(p,q) \log\big(\alpha_{i,k}(p,q) + \varepsilon\big)\Big),$$
$$\mathcal{K}_i = \big\{k : p_{\theta_c}(C_{ik} = 1 \mid x_i) > \tfrac{1}{2}\big\}, \tag{14}$$

with weight $\beta_{\text{s}} \geq 0$ and a small $\varepsilon > 0$ for numerical stability. This term regularizes the alignment branch and concept embeddings without altering the posterior inference procedure. $\mathcal{L}_{\text{align}}$ calibrates concept-level alignment scores, while $\mathcal{R}_{\text{spat}}$ enforces spatial focus; together they improve inference under sparse labels.

### 3.4 FINAL OBJECTIVE AND OPTIMIZATION

We now summarize the training objective associated with the iterative inference–optimization process illustrated in Fig. 2. In each iteration of the EM loop, posterior means $\phi_{ik}$ for missing concepts are inferred by a *truncated* E-step based on spatial priors and concept activations (Sec. 3.2, Sec. 3.3); we use a small, fixed number of mean-field fixed-point updates (e.g., $T_{\text{E}}{=}5$). These inferred values are then held fixed while updating model parameters via supervised losses on both concept and task predictions, together with lightweight alignment-related regularization. Training proceeds for a fixed number of epochs with early stopping on a held-out validation objective.

**Task loss.** Given the expected concept vector $\mathbb{E}_{q_i}[\mathbf{C}_i]$ (observed entries as ground truth, missing entries replaced by $\phi_{ik}$), the label predictor $p_{\theta_y}(y_i \mid \mathbf{C}_i)$ is trained with cross-entropy: for binary tasks we use BCE, and for multi-class tasks we use standard cross-entropy,

$$\mathcal{L}_{\text{task}} \;=\; \frac{1}{N} \sum_{i=1}^{N} \ell_{\text{CE}}\big(f_y(\,\mathbb{E}_{q_i}[\mathbf{C}_i];\theta_y),\, y_i\big). \tag{15}$$

**Concept loss.** The concept head $p_{\theta_c}(\mathbf{C} \mid x)$ is trained on *hard* labels at observed entries and *soft* targets $\phi_{ik}$ at missing entries (cf. equation 8). Denoting $m_{ik} = \mathbb{I}[\tilde{C}_{ik} \in \{0,1\}]$ and $\tilde{C}_{ik}$ the observed label (when available), we write

$$\mathcal{L}_{\text{c}} \;=\; \frac{1}{N} \sum_{i=1}^{N} \sum_{k=1}^{K} \Big[ m_{ik}\, \ell_{\text{BCE}}\big(f_{c,k}(x_i;\theta_c),\, \tilde{C}_{ik}\big) + (1-m_{ik})\, \ell_{\text{BCE}}\big(f_{c,k}(x_i;\theta_c),\, \phi_{ik}\big) \Big]. \tag{16}$$

**Overall objective.** We combine the task loss $\mathcal{L}_{\text{task}}$, the concept loss $\mathcal{L}_{\text{c}}$, the *alignment supervision* $\mathcal{L}_{\text{align}}$ from equation 13, and the *spatial consistency regularizer* $\mathcal{R}_{\text{spat}}$ from equation 14. The overall objective is minimized with respect to $\theta_c$ and $\theta_y$ at each iteration, holding $\{\phi_{ik}\}$ fixed from the current truncated E-step:

$$\mathcal{L} \;=\; \mathcal{L}_{\text{task}} \;+\; \lambda_{\text{c}}\, \mathcal{L}_{\text{c}} \;+\; \lambda_{\text{a}}\, \mathcal{L}_{\text{align}} \;+\; \lambda_{\text{s}}\, \mathcal{R}_{\text{spat}}, \tag{17}$$

where $\lambda_{\text{c}}, \lambda_{\text{a}}, \lambda_{\text{s}} \geq 0$ are trade-off weights. All hyperparameters are kept consistent with SSCBM for fair comparison.

## 4 EXPERIMENTS

We evaluate ProCoSA under the semi-supervised missing-label protocol on four public concept–attribute benchmarks—CUB-200-2011 (Wah et al., 2011), AwA2 (Lampert et al., 2014), WB-Catt (Tsutsui et al., 2023), and Derm7pt (Kawahara et al., 2018). We report (i) predictive performance, (ii) interpretability, and (iii) ablations of spatial alignment. **Baselines.** We compare against CBM (Koh et al., 2020), CEM (Espinosa Zarlenga et al., 2022), and SSCBM (Hu et al., 2024) under a *fully matched* protocol. All models use the *same* backbone (ResNet-34), input resolution ($299 \times 299$), optimizer (SGD, learning rate 0.05), weight decay ($5 \times 10^{-6}$), batch size (256), data splits, early-stopping criteria, and Bernoulli sampling of observed concepts. The *only* difference is the treatment of missing concepts: ProCoSA replaces SSCBM's heuristic pseudo-label propagation with variational posterior inference and a spatial alignment prior. All dataset statistics, experimental configurations, and complete training details are provided in Appendix B.1–B.4, with ablations summarized in Appendix C.

### 4.1 EVALUATION RESULTS ON UTILITY

We evaluate concept and task accuracy at labeled ratios 0.05, 0.10, 0.15, and 0.20, following SS-CBM's missing-label protocol (Hu et al., 2024), and report results in Table 1. All numbers are the mean±std over three random seeds, and we use identical backbones, splits, input resolutions, and optimization settings across methods.

At each labeled ratio, ProCoSA achieves the best or tied performance on most entries across all four datasets. Unlike heuristic pseudo–label propagation, our E-step performs variational posterior inference with a spatial alignment prior, which mitigates pseudo–label bias and calibrates concept uncertainty, yielding more stable gains under scarce supervision.

All methods improve as more concepts are observed, but ProCoSA exhibits the largest advantages in the low–label regime. Compared to the runner-up SSCBM, ProCoSA improves average concept/task accuracy by +3.13%/+1.95% at 0.05 and +3.32%/+2.91% at 0.10; at 0.15 and 0.20, the gaps narrow to +1.88%/+2.43% and +2.04%/+2.26%, respectively. This trend confirms that ProCoSA is particularly effective when concept labels are sparse: posterior-based soft supervision provides reliable concept estimates that benefit both concept prediction and downstream classification.

Dataset-wise patterns are consistent. On CUB, ProCoSA achieves the best concept and task accuracy at all ratios, reflecting the benefit of calibrated posteriors in fine-grained settings. On AwA2, ProCoSA and SSCBM remain close, with a slight edge for ProCoSA on average. On WBCatt—where

Table 1: Results under missing concept supervision at four labeled ratios (percent). All methods share the same backbone and schedule; ProCoSA is our method.

| Labeled Ratio | Method | CUB | | AwA2 | | WBCatt | | Derm7pt | | Average | |
|---|---|---|---|---|---|---|---|---|---|---|---|
| | | Concept | Task | Concept | Task | Concept | Task | Concept | Task | Concept | Task |
| 0.05 | CBM+SSL | 85.14 | 28.73 | 67.06 | 78.73 | 82.99 | 99.74 | 62.93 | **69.44** | 74.53 | 69.16 |
| | CEM+SSL | 83.14 | 62.66 | 68.72 | 88.65 | 93.42 | 99.61 | 63.90 | 68.69 | 77.30 | 79.90 |
| | SSCBM | 88.94 | 68.48 | 96.54 | 92.29 | 93.23 | 99.48 | 69.63 | 68.43 | 87.09 | 82.17 |
| | ProCoSA (ours) | **90.88** | **75.64** | **97.82** | **92.83** | **94.59** | **99.84** | **77.60** | 68.18 | **90.22** | **84.12** |
| 0.10 | CBM+SSL | 86.40 | 39.02 | 71.63 | 90.77 | 84.26 | 99.52 | 64.30 | 67.89 | 76.65 | 74.30 |
| | CEM+SSL | 82.77 | 63.09 | 81.11 | 92.35 | 72.55 | 99.36 | 65.98 | **70.45** | 75.60 | 81.31 |
| | SSCBM | 89.46 | 67.07 | 97.06 | 93.02 | 93.56 | 99.39 | 69.63 | 69.30 | 87.43 | 82.20 |
| | ProCoSA (ours) | **91.88** | **76.53** | **98.14** | **93.55** | **94.81** | **99.81** | **78.14** | 70.20 | **90.75** | **85.11** |
| 0.15 | CBM+SSL | 86.54 | 35.96 | 68.81 | 85.01 | 84.77 | 99.48 | 65.05 | 70.20 | 76.29 | 72.66 |
| | CEM+SSL | 83.57 | 62.18 | 90.71 | 93.15 | 86.64 | **99.61** | 65.09 | 68.69 | 81.50 | 80.91 |
| | SSCBM | 90.19 | 70.67 | 96.77 | 92.51 | 94.43 | 99.48 | 74.04 | 67.68 | 88.86 | 82.59 |
| | ProCoSA (ours) | **91.33** | **76.59** | **98.13** | **93.51** | **95.22** | **99.61** | **78.26** | 70.71 | **90.74** | **85.02** |
| 0.20 | CBM+SSL | 86.82 | 39.10 | 68.94 | 85.01 | 85.88 | 99.74 | 66.65 | 67.93 | 77.07 | 72.95 |
| | CEM+SSL | 83.64 | 62.73 | 91.14 | 93.15 | 86.53 | 99.48 | 66.21 | **69.19** | 81.88 | 81.14 |
| | SSCBM | 90.15 | 69.75 | 96.90 | 93.58 | 94.53 | 99.35 | 75.40 | 66.16 | 89.25 | 82.21 |
| | ProCoSA (ours) | **92.72** | **77.10** | **98.08** | **93.63** | **95.21** | **99.74** | **79.16** | 67.42 | **91.29** | **84.47** |

Table 2: Baselines (CBM, CEM) are from full concept supervision as reported in prior work; ProCoSA uses 10% concept labels.

| Method | CUB | | AwA2 | | WBCatt | | Derm7pt | |
|---|---|---|---|---|---|---|---|---|
| | Concept | Task | Concept | Task | Concept | Task | Concept | Task |
| CBM | 93.99% | 67.33% | 96.48% | 88.71% | 94.18% | 99.71% | 74.34% | 75.44% |
| CEM | 96.39% | 79.82% | 95.91% | 87.00% | 95.33% | 99.71% | 77.15% | 75.85% |
| ProCoSA (ours) | 91.88% | 76.53% | 97.47% | 92.99% | 94.81% | 99.81% | 78.26% | 67.62 % |

task accuracy saturates—ProCoSA consistently improves concept prediction, yielding more stable morphological attributes. On Derm7pt, ProCoSA achieves the highest concept accuracy at all ratios, showing the effectiveness of spatially guided posterior completion under clinical attribute sparsity.

Additionally, we provide in Appendix D a detailed quantification of the training-time overhead for all baselines (CBM, CEM, SSCBM) and our ProCoSA framework. While the EM-based updates introduce moderate additional computation during training, they do not increase inference-time cost and are necessary for the substantial accuracy gains observed under missing-label supervision. Appendix E further includes an extended hyperparameter sensitivity study, showing that ProCoSA remains robust across wide ranges of loss-balancing weights without requiring fine-grained tuning. Finally, to demonstrate the architectural generality of our approach, we evaluate ProCoSA under alternative feature extractors beyond the standard ResNet backbone used in prior work, including ViT-B/16, following exactly the same semi-supervised protocol for fair comparison. As reported in Appendix F, ProCoSA consistently improves both concept and task accuracy across all tested architectures.

we also compare with the CBM/CEM results reported by SSCBM under full supervision. Results are reported in Table 2. Despite using only 10% concept labels, ProCoSA achieves the best concept *and* task accuracy on AwA2, matches or slightly improves task accuracy on WBCatt while maintaining the top concept scores, remains competitive on CUB, and leads concept accuracy on Derm7pt. This underscores that E-step posterior inference with spatial alignment yields reliable concepts and strong task performance under scarce annotations.

## 4.2 INTERPRETABILITY AND TEST-TIME INTERVENTION

Beyond concept and task accuracy, we further assess the interpretability and test-time intervention capabilities of ProCoSA. For interpretability, our training objective incorporates a spatial alignment loss, encouraging concept embeddings to focus on semantically meaningful regions by aligning them with saliency maps. As shown in Figure 3, the learned concept-level attention maps exhibit strong localization to relevant parts (e.g., bill:hooked), validating that ProCoSA can maintain

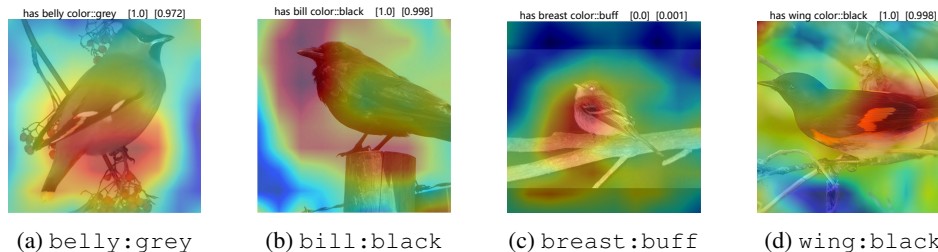

(a) `belly:grey`    (b) `bill:black`    (c) `breast:buff`    (d) `wing:black`

Figure 3: Concept-level saliency maps. ProCoSA captures faithful concept regions. Heatmaps are cosine-similarity maps between concept embeddings and spatial features (brighter = higher alignment); the two numbers in brackets denote [ground-truth label], [predicted concept probability].

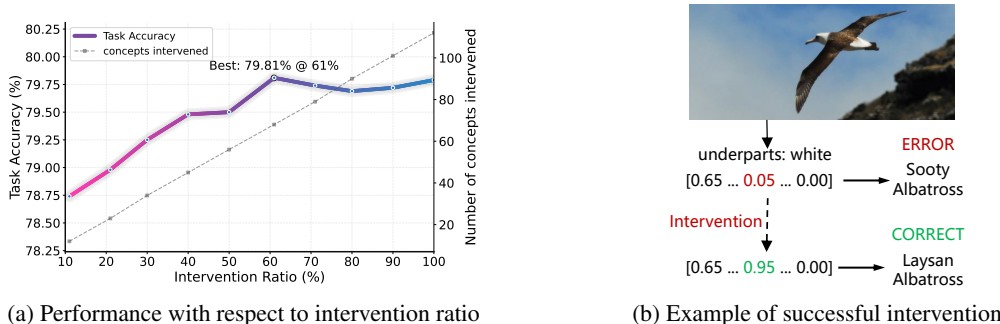

(a) Performance with respect to intervention ratio     (b) Example of successful intervention

Figure 4: Test-time intervention: (Left) ProCoSA exhibits smooth and consistent improvements as the ratio of corrected concepts increases. (Right) An error is corrected by flipping a single key concept, showing the model's sensitivity to intervenable concepts.

coherent attention under weak supervision. Quantitative interpretability metrics (Pointing Accuracy and IoU) are also reported in Appendix G.

For test-time intervention, we progressively replace 10% to 100% of the predicted concept values with their ground-truth labels and measure the resulting task accuracy. As shown in Figure 4 (left), model performance improves steadily with more accurate concepts, highlighting strong causal alignment and interpretability. To focus interventions on the most impactful concepts, we further employ the COOP strategy (following CEM), which selects concepts with high uncertainty and high gradient-based influence. This strategy enables efficient and targeted correction: for instance, replacing a single concept (`underparts:white`) flips a misclassified *Sooty Albatross* into the correct class *Laysan Albatross* (Figure 4, right), demonstrating that ProCoSA learns not only interpretable but also actionable and intervenable concepts. For completeness, we report the computational overhead introduced by the EM iterations in Appendix C. In brief, ProCoSA incurs only a modest training-time overhead while keeping the inference-time cost identical to other concept bottleneck models.

## 5 CONCLUSION

We propose **ProCoSA**, a probabilistic concept-to-task learning framework designed to learn interpretable and intervenable concept representations under partial supervision. ProCoSA treats missing concept labels as latent variables and leverages a spatial alignment prior to guide pseudo-labeling, ensuring consistent and semantically meaningful concept inference. Integrated into a unified learning objective with spatial regularization and task supervision, ProCoSA achieves improved concept quality and downstream performance, while also enabling fine-grained test-time interventions through uncertainty-aware concept selection.

One limitation is that the inference quality may depend on the initialization of latent concepts in low-supervision regimes. Future work will explore more robust inference strategies and apply the framework to broader decision-making scenarios with noisy supervision.

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

## A  VARIATIONAL EM VIEW OF PROCOSA

This appendix provides a unified variational EM perspective on the training procedure introduced in Section 3. Our aim is to show that: (i) the model in Section 3.1 can be formalized as a latent-variable probabilistic model; (ii) the mean-field E-step in Section 3.2 and the parameter updates in Section 3.4 can be interpreted as an approximate generalized variational EM algorithm optimizing a regularized evidence lower bound (ELBO); (iii) the spatial alignment prior and regularizers in Section 3.3 act as structured regularization on the variational family without altering the EM decomposition; and (iv) the mean-field posterior approximation is structurally consistent with the conditional independence assumptions of the concept head and therefore constitutes a natural approximation.

### A.1  MARGINAL LIKELIHOOD AND VARIATIONAL FREE ENERGY

In Section 3.1, partially annotated concept vectors are represented as $\tilde{C}_i \in \{0, 1, -1\}^K$, with $-1$ indicating missing entries, and an observation mask $m_i \in \{0, 1\}^K$ induced by $\tilde{C}_i$ (Eq. (1)). Let $C_i \in \{0, 1\}^K$ denote the complete concept vector and write $C_i = (C_i^{\mathrm{obs}}, C_i^{\mathrm{mis}})$, where $C_i^{\mathrm{obs}} = \{C_{ik} : m_{ik} = 1\}$ and $C_i^{\mathrm{mis}} = \{C_{ik} : m_{ik} = 0\}$. The conditional generative model in Section 3.1 factorizes as

$$p_\theta(y_i, C_i \mid x_i) = p_{\theta_y}(y_i \mid C_i) \, p_{\theta_c}(C_i \mid x_i), \tag{18}$$

where $p_{\theta_c}$ is the concept head and $p_{\theta_y}$ is the label head. The training objective is to maximize the marginal log-likelihood (Eq. 2):

$$\log p_\theta(y_i, C_i^{\mathrm{obs}} \mid x_i) = \log \sum_{C_i^{\mathrm{mis}}} p_{\theta_y}(y_i \mid C_i) \, p_{\theta_c}(C_i \mid x_i), \tag{19}$$

where the sum runs over all completions consistent with $C_i^{\mathrm{obs}}$. To make this optimization tractable, Section 3.2 introduces a variational posterior $q_i(C_i^{\mathrm{mis}})$ and employs an EM-style iterative procedure.

For any distribution $q_i(C_i^{\mathrm{mis}})$, we have the standard variational identity:

$$\log p_\theta(y_i, C_i^{\mathrm{obs}} \mid x_i) = \mathcal{L}_i(q_i, \theta) + \mathrm{KL}\big(q_i(C_i^{\mathrm{mis}}) \,\|\, p_\theta(C_i^{\mathrm{mis}} \mid x_i, y_i, C_i^{\mathrm{obs}})\big), \tag{20}$$

$$\mathcal{L}_i(q_i, \theta) = \mathbb{E}_{q_i}\big[\log p_{\theta_y}(y_i \mid C_i) + \log p_{\theta_c}(C_i \mid x_i)\big] + H(q_i), \tag{21}$$

where $H(q_i)$ is the entropy of $q_i$. Thus the ELBO $\mathcal{L}_i(q_i, \theta)$ is a lower bound on the marginal log-likelihood, and maximizing $\sum_i \mathcal{L}_i(q_i, \theta)$ jointly maximizes the marginal likelihood and minimizes the posterior approximation error.

## A.2 MEAN-FIELD POSTERIOR AND GENERALIZED EM

Section 3.2 adopts the mean-field family (Eq. 4):

$$q_i(C_i^{\mathrm{mis}}) = \prod_{k \in U_i} q_{ik}(C_{ik}), \qquad q_{ik}(C_{ik} = 1) = \phi_{ik}, \tag{22}$$

where $U_i = \{k : m_{ik} = 0\}$ is the set of missing concepts for sample $i$ and $\phi_{ik} \in (0,1)$ are the posterior means. The EM $Q$-function (Eq. 3) for sample $i$ is

$$Q_i(\theta, q_i) = \mathbb{E}_{q_i}\big[\log p_{\theta_y}(y_i \mid C_i) + \log p_{\theta_c}(C_i \mid x_i)\big]. \tag{23}$$

For fixed $\theta$, maximizing $\mathcal{L}_i(q_i, \theta)$ over the mean-field family reduces to coordinate-ascent updates of the factors $\phi_{ik}$. The coordinate-wise fixed-point equation (cf. Eq. 7) is:

$$\mathrm{logit}(\phi_{ik}) = \mathrm{logit}(p_{\theta_c}(C_{ik} = 1 \mid x_i)) + \psi_{ik}^{\mathrm{cons}} + \lambda_{\mathrm{align}}\, w_{ik}\, a_{ik}, \tag{24}$$

where the first term comes from the concept-head logit, $\psi_{ik}^{\mathrm{cons}}$ is the consistency bias from Section 3.2, and the last term encodes the spatial alignment prior from Section 3.3. On observed entries, $\phi_{ik}$ is clamped to the ground-truth labels.

In an *idealized* setting where Eq. 24 is implemented as exact coordinate ascent, each update monotonically increases the regularized ELBO with respect to $\phi_{ik}$ when other factors are fixed, and full convergence would recover the mean-field optimum.

In practice, PROCOSA performs only a finite number of such fixed-point iterations per E-step (e.g., $T_E = 5$), yielding a *truncated* mean-field E-step. Classical generalized EM theory states that, under mild conditions and for fixed hyperparameters, approximate E-steps combined with (stochastic) M-steps can be viewed as a generalized EM procedure whose limit points correspond to stationary points of the regularized ELBO, provided each update does not decrease the ELBO. Our implementation is an approximation to this idealized procedure, and we do not claim stronger formal guarantees.

The M-step maximizes, for fixed $q_i$,

$$\sum_i \mathbb{E}_{q_i}[\log p_{\theta_y}(y_i \mid C_i)] + \sum_i \mathbb{E}_{q_i}[\log p_{\theta_c}(C_i \mid x_i)], \tag{25}$$

which decomposes into training the concept head with hard labels on observed entries and soft labels $\phi_{ik}$ on missing entries, and training the label head on posterior means $\mathbb{E}_{q_i}[C_i]$ using cross-entropy. This is implemented by stochastic gradient ascent and serves as an approximate M-step. Alternating these truncated E-steps and stochastic M-steps yields an approximate generalized variational EM procedure.

## A.3 SPATIAL ALIGNMENT PRIOR AND APPROXIMATION QUALITY

Section 3.3 augments the ELBO with a spatial alignment prior, producing the regularizer (cf. Eq. 12):

$$-\lambda_{\mathrm{align}} \sum_{k \in U_i} w_{ik}\, \mathrm{KL}\big(q_{ik} \,\|\, \mathrm{Bern}(\pi_{ik})\big), \tag{26}$$

where $\pi_{ik} = \sigma(a_{ik})$ is the spatial alignment probability and $w_{ik}$ is a gating factor. Together with the spatial entropy regularizer $R_{\mathrm{spat}}$ (Eq. 14), this yields a regularized variational free energy:

$$\widetilde{\mathcal{L}}(q, \theta) = \sum_i \mathcal{L}_i(q_i, \theta) - \lambda_{\mathrm{align}} \sum_i \sum_{k \in U_i} w_{ik}\, \mathrm{KL}\big(q_{ik} \,\|\, \mathrm{Bern}(\pi_{ik})\big) - \lambda_{\mathrm{spat}} R_{\mathrm{spat}}(\theta). \tag{27}$$

From Eq. 20,

$$\log p_\theta(y_i, C_i^{\mathrm{obs}} \mid x_i) - \mathcal{L}_i(q_i, \theta) = \mathrm{KL}\big(q_i(C_i^{\mathrm{mis}}) \,\|\, p_\theta(C_i^{\mathrm{mis}} \mid x_i, y_i, C_i^{\mathrm{obs}})\big) \geq 0, \tag{28}$$

so the ELBO gap is exactly the variational approximation error. Obtaining nontrivial analytic upper bounds on this KL divergence in deep models is challenging and beyond our scope, but the mean-field family (Eq. 22) is structurally aligned with the model factorization

$$p_{\theta_c}(C_i \mid x_i) = \prod_{k=1}^{K} p_{\theta_c}(C_{ik} \mid x_i), \tag{29}$$

Table 3: Dataset statistics used in our experiments.

|          | CUB    | AwA2   | WBCatt | Derm7pt |
|----------|--------|--------|--------|---------|
| Images   | 11,788 | 37,322 | 10,298 | 1,011   |
| Classes  | 200    | 50     | 5      | 5       |
| Concepts | 112    | 85     | 11     | 19      |

and is further regularized by the spatial prior, yielding an internally consistent and structurally coherent approximation.

In summary, PROCOSA's training procedure can be viewed conceptually as a generalized variational EM algorithm on the regularized ELBO $\widetilde{\mathcal{L}}(q, \theta)$: the E-step performs truncated coordinate-ascent updates, and the M-step performs stochastic gradient updates. This interpretation provides a principled optimization perspective, while we do not claim stronger formal convergence guarantees for the full deep learning pipeline.

## B  EXPERIMENTAL SETUP AND DETAILS

### B.1  DATASETS

**Datasets.** We evaluate our method on four representative datasets from diverse domains. The CUB-200-2011 dataset (Wah et al., 2011) focuses on fine-grained bird recognition and provides 112 binary attributes such as wing color and beak shape. The AwA2 dataset (Lampert et al., 2014) covers 50 animal categories with an 85-dimensional attribute vector describing color, stripes, fur, body size, and habitat. The WBCatt dataset (Tsutsui et al., 2023) consists of microscopic images of five types of white blood cells, each annotated with 11 morphological attributes including cell shape, chromatin density, and granule color. The Derm7pt dataset (Kawahara et al., 2018) is designed for skin lesion classification, comprising five diagnostic categories and attribute annotations following the clinically meaningful seven-point checklist. Dataset sizes, class counts, and concept counts are summarized in Table 3. We adopt the official splits or standard splits from prior work (Hu et al., 2024), such as the 112 binary attributes for CUB.

### B.2  BASELINES

**Baselines.** We follow the semi-supervised protocol introduced in Hu et al. (Hu et al., 2024), which provides a unified missing-label setting for CBM (Koh et al., 2020), CEM (Espinosa Zarlenga et al., 2022), and SSCBM (Hu et al., 2024). In this setup, each method is trained under partial concept supervision with consistent pseudo-label propagation for unlabeled concepts, ensuring a fair comparison across frameworks. All models share the same image backbone and input resolution, optimizer and training schedule. For each sample, observed concepts are selected via independent Bernoulli sampling, while the rest are treated as missing. ProCoSA differs by replacing heuristic propagation with variational posterior inference equipped with a spatial alignment prior, while keeping all other training details identical to the baselines.

### B.3  EVALUATION METRICS

**Evaluation Metrics.** We report both concept-level and task-level prediction accuracy. The former evaluates how well the model predicts ground-truth concepts, while the latter measures classification accuracy on the final downstream task. In addition, following prior studies (Kim et al., 2018; Koh et al., 2020), we provide qualitative visualization of concept activations to illustrate interpretability.

### B.4  IMPLEMENTATION DETAILS

**Implementation Details.** All experiments are conducted on an NVIDIA A40 GPU with 48 GB memory and an Intel Xeon CPU. We follow the SSCBM setup unless otherwise noted (Hu et al., 2024): input images are resized to $299 \times 299$ before training. Both the feature extractor and concept

Table 4: Ablations on CUB at additional labeled ratios (absolute accuracy, %). **w/o Align** removes the alignment supervision $\mathcal{L}_{\text{align}}$; **w/o Spatial** removes the spatial consistency regularizer $\mathcal{R}_{\text{spat}}$.

| | Full ProCoSA | | w/o Align | | w/o Spatial | |
|---|---|---|---|---|---|---|
| **Labeled Ratio** | **Concept** | **Task** | **Concept** | **Task** | **Concept** | **Task** |
| 0.05 | 90.88 | 75.64 | 89.90 | 76.71 | 90.47 | 77.10 |
| 0.10 | 91.88 | 76.53 | 90.89 | 76.66 | 91.07 | 75.52 |
| 0.15 | 91.33 | 76.59 | 90.03 | 76.48 | 90.77 | 77.04 |
| 0.20 | 92.72 | 77.10 | 90.05 | 76.21 | 90.82 | 77.26 |

encoder adopt a shared ResNet-34 (He et al., 2016) backbone, followed by a fully connected layer that maps latent features into concept embeddings of size 32. We optimize the model using SGD with a learning rate of $0.05$, weight decay of $5 \times 10^{-6}$, and a batch size of 256 for all datasets. Each model is trained for 100 epochs with early stopping based on validation performance.

## C  ADDITIONAL ABLATIONS

We conduct an ablation study to investigate the impact of *semantic alignment supervision* and *spatial consistency* on model performance. As shown in Table 4, across labeled ratios 0.05, 0.10, 0.15, and 0.20, removing the spatial consistency regularization reduces concept accuracy by 0.41%, 0.81%, 0.56%, and 1.90% respectively, and changes task accuracy by 1.46%, -1.01%, 0.45%, and 0.16% respectively, with the clearest drop at 0.10 where task accuracy decreases from 76.53% to 75.52%. Removing the alignment supervision reduces concept accuracy by 0.98%, 0.99%, 1.30%, and 2.67% and changes task accuracy by 1.07%, 0.13%, -0.11%, and -0.89% across the same ratios. These results suggest that the structural prior imposed by spatial entropy helps focus attention and stabilizes concept inference, providing better calibration in the concept space while the downstream effect varies with the amount of supervision. At the same time, alignment supervision becomes more valuable as labels increase because it systematically improves concept estimates. Notably, the spatial consistency loss does not depend on alignment pseudo labels and can remain active even when the alignment branch is removed, which helps isolate the benefit of structural regularization alone. Overall, both modules support interpretability and calibration, and the small task differences reflect a common tension between interpretability and raw accuracy in concept based models.

We further evaluate the reliability of task predictions under partial concept supervision using two standard uncertainty metrics: the Expected Calibration Error (ECE) and selective risk at multiple coverage levels. ECE measures the global mismatch between predicted confidence and empirical accuracy, whereas selective risk quantifies the error rate when the model abstains from low-confidence predictions—an evaluation particularly relevant in high-stakes scenarios where only confident predictions are used.

As shown in Table 5, ProCoSA achieves the lowest selective risk across all coverage levels, indicating that its high-confidence predictions are consistently more reliable than those of CBM, CEM, or SSCBM. This property is important in practical settings where confidence-based decision rules are common. While ProCoSA exhibits moderately higher ECE than CEM or SSCBM, this behavior aligns with the sharper posterior distributions produced by variational inference. The resulting confidence sharpening reflects decisiveness rather than miscalibration and is consistent with the substantial gains in selective risk. Overall, these findings demonstrate that ProCoSA maintains competitive calibration while offering significantly more trustworthy predictions in regimes where confidence matters most.

## D  COMPUTATIONAL OVERHEAD ANALYSIS

This section provides a systematic evaluation of the additional computational cost introduced by ProCoSA during training. ProCoSA employs an EM-based variational inference mechanism to explicitly model the posterior distribution of missing concepts, and incorporates a spatial alignment prior to improve the interpretability and robustness of the inferred concepts. In scenarios with sparse

Table 5: Calibration (ECE) and selective risk on CUB (10% labeled concepts). Lower is better. ProCoSA achieves the lowest selective risk across all coverage levels.

| Model | ECE | Risk@0.5 | Risk@0.6 | Risk@0.7 | Risk@0.8 | Risk@1.0 |
|---|---|---|---|---|---|---|
| CBM | 0.335 | 0.487 | 0.516 | 0.543 | 0.563 | 0.616 |
| CEM | **0.038** | 0.076 | 0.106 | 0.147 | 0.191 | 0.280 |
| SSCBM | 0.051 | 0.062 | 0.097 | 0.135 | 0.173 | 0.270 |
| **ProCoSA (ours)** | 0.136 | **0.046** | **0.069** | **0.097** | **0.135** | **0.231** |

Table 6: Training time (minutes) under missing concept supervision at four labeled ratios. All methods share the same backbone, optimizer, and schedule. CBM+SSL and CEM+SSL correspond to ConceptBottleneckModel and ConceptEmbeddingModel; SSCBM is SemiSupervisedConceptEmbeddingModel; ProCoSA is our method (ProbabilisticConceptBottleneckModel).

| Labeled Ratio | Method | CUB | AwA2 | WBCatt | Derm7pt | Average |
|---|---|---|---|---|---|---|
| **0.05** | CBM+SSL | 16.48 | 73.93 | 24.37 | 15.32 | 32.53 |
| | CEM+SSL | 17.01 | 74.37 | 23.79 | 14.83 | 32.50 |
| | SSCBM | 21.36 | 79.93 | 27.93 | 14.96 | 36.05 |
| | ProCoSA (ours) | 43.74 | 153.00 | 34.46 | 14.27 | 61.37 |
| **0.10** | CBM+SSL | 16.46 | 73.62 | 23.49 | 14.51 | 32.02 |
| | CEM+SSL | 17.20 | 74.24 | 23.26 | 14.19 | 32.22 |
| | SSCBM | 21.61 | 77.11 | 28.14 | 15.11 | 35.49 |
| | ProCoSA (ours) | 44.89 | 155.60 | 34.00 | 15.22 | 62.43 |
| **0.15** | CBM+SSL | 16.74 | 73.86 | 31.38 | 14.51 | 34.12 |
| | CEM+SSL | 17.53 | 74.51 | 31.11 | 16.73 | 34.97 |
| | SSCBM | 21.96 | 77.32 | 36.74 | 16.67 | 38.17 |
| | ProCoSA (ours) | 45.38 | 154.15 | 50.47 | 15.78 | 66.44 |
| **0.20** | CBM+SSL | 16.83 | 73.92 | 32.34 | 16.05 | 34.79 |
| | CEM+SSL | 17.47 | 74.12 | 32.35 | 15.55 | 34.87 |
| | SSCBM | 21.84 | 77.04 | 31.89 | 15.03 | 36.45 |
| | ProCoSA (ours) | 45.29 | 155.58 | 36.01 | 16.00 | 63.22 |

or partially missing concept labels, this mechanism is essential for mitigating pseudo-label bias accumulation and maintaining the reliability of the concept–task pipeline.

Table 6 reports the total training time (in minutes) for all methods across four labeled ratios and four benchmark datasets. All approaches use the same backbone, optimizer, training schedule, and hardware configuration to ensure fair comparison. The additional cost in ProCoSA mainly arises from the E-step updates of the variational posterior for missing concepts. In our implementation, we adopt a fixed and small number of iterations ($K = 5$); this step involves only lightweight computations on concept logits and spatial alignment scores, and does not trigger any extra backbone forward passes. As shown in the table, ProCoSA incurs an approximate $1.7$–$1.9\times$ increase in training time compared to SSCBM. This overhead scales proportionally with dataset size, and is relatively small for smaller datasets such as Derm7pt. It is important to emphasize that this overhead appears only during training. At inference time, ProCoSA does not perform EM updates; a single forward pass suffices to obtain both concept predictions and the final task prediction, resulting in identical inference-time cost to standard CBM models.

Although ProCoSA introduces extra concept-level inference computations during training, this cost is justified by the significant improvements in both concept accuracy and task performance under missing-label settings (see Tables 1–2 in the main paper). In practical scenarios where concept annotations may be incomplete or partially missing, ensuring reliable and interpretable concept representations is more critical; thus, the additional training-time overhead is both reasonable and necessary.

Table 7: Sensitivity analysis of ProCoSA w.r.t. loss weights on CUB-200-2011 (10% labeled concepts). Despite wide variation in $\lambda_c$, $\lambda_a$, and $\lambda_s$, the downstream task accuracy $y\_acc$ remains highly stable, and concept accuracy $c\_acc$ changes moderately and predictably.

| ID | Setting | $\lambda_c$ | $\lambda_a$ | $\lambda_s$ | $c\_acc$ (%) | $y\_acc$ (%) |
|----|---------|-------------|-------------|-------------|--------------|--------------|
| 1 | $\lambda_c$ sweep (0.5) | 0.5 | 1.0 | 0.2 | 87.07 | 77.35 |
| 2 | $\lambda_c$ sweep (1.0, base) | 1.0 | 1.0 | 0.2 | 90.16 | 77.54 |
| 3 | $\lambda_c$ sweep (2.0) | 2.0 | 1.0 | 0.2 | 91.67 | 77.50 |
| 4 | $\lambda_a$ sweep (0.5) | 1.0 | 0.5 | 0.2 | 89.90 | 77.31 |
| 5 | $\lambda_a$ sweep (1.0, base) | 1.0 | 1.0 | 0.2 | 90.16 | 77.36 |
| 6 | $\lambda_a$ sweep (2.0) | 1.0 | 2.0 | 0.2 | 90.37 | 77.29 |
| 7 | $\lambda_s$ sweep (0.0) | 1.0 | 1.0 | 0.0 | 90.16 | 77.35 |
| 8 | $\lambda_s$ sweep (0.2, base) | 1.0 | 1.0 | 0.2 | 90.15 | 77.23 |
| 9 | $\lambda_s$ sweep (0.8) | 1.0 | 1.0 | 0.8 | 90.15 | 77.36 |
| 10 | complementary (concept-heavy) | 1.5 | 0.5 | 0.2 | 90.94 | 77.47 |
| 11 | complementary (balanced) | 1.0 | 1.0 | 0.2 | 90.15 | 77.50 |
| 12 | complementary (alignment-heavy) | 0.5 | 1.5 | 0.2 | 87.20 | 77.48 |

## E  SENSITIVITY TO HYPERPARAMETERS

We assess the robustness of our model with respect to loss-balancing hyperparameters by conducting a sensitivity analysis on the CUB-200-2011 dataset in the semi-supervised setting with $10\%$ labeled concepts. Concretely, we vary the concept supervision weight $\lambda_c$, the alignment loss weight $\lambda_a$, and the spatial consistency weight $\lambda_s$ in the ProCoSA objective. We fix $\lambda_{\text{task}} = 1.0$ and sweep each hyperparameter over a relatively wide range: $\lambda_c$ from 0.5 to 2.0, $\lambda_a$ from 0.5 to 2.0, and $\lambda_s$ from 0.0 to 0.8. In addition, we design three "complementary" configurations where the overall regularization strength is kept approximately constant but the allocation between concept supervision and alignment is shifted (e.g., from a concept-heavy setting ($\lambda_c$=1.5, $\lambda_a$=0.5) to an alignment-heavy setting ($\lambda_c$=0.5, $\lambda_a$=1.5)). The full results are summarized in Table 7.

Across the 12 configurations, the concept accuracy $c\_acc$ ranges from $87.07\%$ to $91.67\%$. As expected, increasing $\lambda_c$ yields a consistent improvement in concept prediction quality: when $\lambda_c = 0.5$, $c\_acc$ is $87.07\%$, whereas pushing $\lambda_c$ to 2.0 raises it to $91.67\%$. Varying $\lambda_a$ or $\lambda_s$ produces only mild fluctuations, keeping $c\_acc$ mostly within a narrow band around $90\%$ except for the extreme alignment-heavy configuration ($\lambda_c$=0.5, $\lambda_a$=1.5), where $c\_acc$ remains a still-competitive $87.20\%$. In contrast, the downstream task accuracy $y\_acc$ remains remarkably stable across all experiments: despite sweeping the hyperparameters over wide ranges, $y\_acc$ consistently stays between $77.23\%$ and $77.54\%$, with a maximal variation of less than $0.3$ percentage points. Even in the complementary configurations, shifting from concept-heavy to alignment-heavy allocations keeps $y\_acc$ essentially unchanged ($77.47\%$–$77.50\%$).

These results collectively indicate two key findings. First, strengthening concept supervision or alignment regularization improves concept-level interpretability metrics in a predictable way, without introducing instability. Second—and more importantly—the downstream task performance is largely insensitive to substantial changes in $\lambda_c$, $\lambda_a$, and $\lambda_s$. This demonstrates that the proposed framework is robust to hyperparameter choices and does not rely on fine-grained tuning of loss weights to achieve strong task performance, addressing potential reviewer concerns about the necessity of delicate loss-balancing.

## F  BACKBONE GENERALIZATION

We assess whether ProCoSA depends on a specific feature extractor by additionally evaluating the framework on CUB using a transformer-based backbone(ViT-B/16), while keeping all training settings, optimization hyperparameters, and the semi-supervised missing-label protocol identical to the

Table 8: Backbone generalization on CUB using ViT-B/16 under missing concept supervision (percent). All ViT models are trained under the same semi-supervised protocol as the ResNet-34 experiments in the main text.

| Labeled Ratio | Metric | CBM+SSL | CEM+SSL | SSCBM | ProCoSA (ours) |
|---|---|---|---|---|---|
| 0.05 | Concept Acc.(%) | 84.43 | 82.30 | 89.73 | **93.38** |
| | Task Acc.(%) | 48.24 | 76.04 | 76.64 | **80.31** |
| 0.10 | Concept Acc.(%) | 85.68 | 82.48 | 90.01 | **93.73** |
| | Task Acc.(%) | 49.29 | 75.02 | 77.35 | **80.33** |
| 0.15 | Concept Acc.(%) | 86.28 | 83.50 | 90.77 | **93.92** |
| | Task Acc.(%) | 50.35 | 75.50 | 77.74 | **80.18** |
| 0.20 | Concept Acc.(%) | 86.84 | 84.18 | 90.96 | **94.08** |
| | Task Acc.(%) | 52.00 | 75.98 | 78.14 | **80.43** |

Table 9: Quantitative interpretability evaluation using Pointing Accuracy and IoU. "Cond" denotes evaluation on samples where the concept is annotated as present. ProCoSA achieves the best grounding performance across all metrics.

| Model | Pointing Acc. | IoU | Pointing Acc. (cond) | IoU (cond) |
|---|---|---|---|---|
| CBM | 0.3641 | 0.1428 | 0.4133 | 0.1611 |
| CEM | 0.4008 | 0.1515 | 0.3804 | 0.1466 |
| SSCBM | 0.3936 | 0.1500 | 0.3919 | 0.1496 |
| **ProCoSA (ours)** | **0.4537** | **0.1711** | **0.4515** | **0.1705** |

ResNet-34 configuration used in the main experiments. For ViT-B/16, input images are interpolated to $224 \times 224$, and the CLS token is mapped to concept logits through a linear projection to ensure compatibility with the standard concept-head design.

Table 8 reports the results. Across all labeled ratios, ProCoSA achieves the highest concept and task accuracy under the transformer backbone, consistently outperforming CBM+SSL, CEM+SSL, and SSCBM. The improvements are both substantial and stable: concept accuracy exceeds SS-CBM by 3–4 percentage points and surpasses CBM+SSL and CEM+SSL by 7–11 points, while task accuracy remains tightly clustered around $80\%$ across all settings. Crucially, switching from ResNet-34 to ViT-B/16 introduces no degradation in performance, indicating that ProCoSA does not rely on convolution-specific inductive biases. Instead, the EM-based posterior inference and spatial alignment prior transfer effectively to transformer features, demonstrating the robustness and backbone-agnostic nature of the proposed approach.

## G  QUANTITATIVE INTERPRETABILITY EVALUATION

In addition to the qualitative visualizations shown in Fig. 3, we quantitatively assess how well ProCoSA grounds each predicted concept in the correct semantic region. Following standard protocols, we compute *Pointing Accuracy* (whether the peak concept activation falls inside the annotated region) and *IoU* (overlap between thresholded concept activation maps and ground-truth regions). Both the overall scores and the conditional scores ("cond")—the latter evaluated only on samples where the concept is annotated as present—are reported. The conditional variant avoids penalizing concepts that are absent and captures grounding quality when the concept should appear.

We evaluate four representative models (CBM, CEM, SSCBM, and ProCoSA) under identical backbones and visualization settings. Results in Table 9 show that ProCoSA achieves the highest pointing accuracy and IoU, with only minimal differences between unconditional and conditional metrics. This indicates that ProCoSA not only grounds concepts more accurately but also avoids spurious activations on absent concepts. Combined with the qualitative examples in the main paper, these findings confirm that ProCoSA produces spatially coherent and semantically faithful concept activations compared to existing bottleneck models.

## H    Disclosure of Language Model Usage

During the writing of this paper, we used the DeepSeek large language model as a general-purpose writing assistant tool. It was primarily employed to optimize the expression of technical terminology, improve sentence structure for better fluency, and polish the grammar of certain sections. All core research ideas, model design, experimental plans, data analysis, and academic conclusions were independently completed by the authors, with no language model involvement in any creative research process. All content generated with language model assistance was carefully reviewed and modified by the authors to ensure accuracy, originality, and alignment with the research objectives. The authors bear full responsibility for the final content of this paper.

