# OpenReview forum: "ProCoSA: Probabilistic Concept Learning with Spatial Alignment"
_ICLR.cc/2026/Conference — ICLR 2026 Conference Withdrawn Submission_

### Official Review · Reviewer_nBqg · 2025-10-29

**Soundness:** 2
**Presentation:** 1
**Contribution:** 2
**Rating:** 2
**Confidence:** 4

**Summary:**

This paper addresses the challenge of learning interpretable concept representations under incomplete or missing concept supervision. To this end, the authors propose ProCoSA, a probabilistic framework that models missing concepts as latent variables and jointly infers concept posteriors and task predictions through an EM procedure. To ensure semantic grounding and reduce pseudo-label bias, ProCoSA introduces a spatial alignment prior that guides inferred concepts to align with salient image regions, supported by lightweight alignment and spatial-consistency regularization. Extensive experiments demonstrate the effectiveness of the proposed ProCoSA.

**Strengths:**

* Compared with previous heuristic pseudo-label propagation approaches (e.g., SSCBM), ProCoSA models concept uncertainty in a Bayesian inference framework, enabling more robust and principled learning of the concept space.
* The authors discuss the related literature in considerable detail.

**Weaknesses:**

1. Interpretability: Although the proposed method shows accuracy improvements over existing approaches, the authors lack sufficient evaluation of the method’s interpretability. Only a few qualitative visualizations are provided. The authors should conduct both qualitative and quantitative analyses to verify that the learned concepts consistently and accurately correspond to the intended semantic regions.
2. Method: The paper uses only ResNet as the feature extraction backbone. The authors should include additional architectures such as ViTs to further demonstrate the generality of the proposed method.
3. Written: This paper is poorly written, with incorrect citation formatting and an unclear presentation of Figure 1.
4. Code: The authors do not provide code for reproducibility check.

**Questions:**

My questions and concerns are in Weaknesses Section.

---

> ### Author Response · Authors · 2025-11-20
>
> We thank the reviewer for the thoughtful comments.
>
> Regarding Question 1, we significantly expanded the interpretability evaluation. Appendix~G (Lines 903--917) now includes Pointing Accuracy and IoU between concept attention and human-annotated regions under a unified protocol. ProCoSA achieves the highest pointing accuracy and competitive IoU across baselines. The main text shows qualitative results and directs readers to the appendix for full quantitative evidence.
>
> Regarding Question 2, we added experiments using a ViT-B/16 backbone in Appendix F (Lines 860--901). ProCoSA continues to outperform CBM, CEM, and SSCBM with transformer features, confirming architecture-independence.
>
> Regarding Question 3, we corrected all citation formatting issues, improved clarity throughout the text, and redrew Figure~1 for clearer structure and flow.
>
> Regarding Question 4, as noted in the revised abstract and per ICLR policy, full code (including preprocessing, training, spatial alignment, and ViT support) will be released immediately upon acceptance to ensure reproducibility.
>
> All revisions have been incorporated into the manuscript. We appreciate the reviewer’s time and hope the improvements are reflected positively in the final evaluation.

---

### Official Review · Reviewer_53yr · 2025-10-31

**Soundness:** 2
**Presentation:** 2
**Contribution:** 2
**Rating:** 4
**Confidence:** 3

**Summary:**

This paper proposes ProCoSA, a method for CBMsthat addresses sparse concept annotations. The core idea is to treat missing concepts as latent variables and infer them using an EM framework. This process is guided by a spatial alignment prior, derived from the cosine similarity between spatial features and concept embeddings, and is supported by two simple regularizers. The model is trained end-to-end, using the posterior mean from the E-step to update the model in the M-step. On four standard CBM datasets, ProCoSA demonstrates improved performance over existing methods in low-label regimes and shows a clear, monotonic improvement in concept intervention curves.

**Strengths:**

1、The paper's use of EM to handle missing labels is a clean, probabilistic alternative to prior heuristic methods like pseudo-labeling.

2、he gating mechanism for the spatial prior is a smart design choice, effectively mitigating noise by applying constraints only when confidence is high.

3、The method shows clear performance gains in low-data regimes and achieves the monotonic intervention curves expected of a well-formed CBM.

**Weaknesses:**

1、Limited Novelty. The paper's core contribution is swapping the k-NN pseudo-labeling from SSCBM with an EM framework. This feels like an incremental methodological refinement rather than a significant conceptual leap.Self-Referential Prior.

2、The spatial alignment prior is not independent, as it shares the same feature backbone with the concept head. This creates a circular problem where a feature is used to generate a prior that in turn constrains itself.

3、The central claim of providing "more reliable posterior uncertainty" is unsubstantiated：1）The paper is missing key experiments on confidence calibration (e.g., ECE, Brier score) to actually prove this；2）the motivation relies on aligning concepts to visual evidence, but the paper lacks any quantitative localization metrics to show this is happening effectively beyond a few qualitative examples.

3）the evaluation is restricted to standard benchmarks,to properly test the method's stability, the analysis should include robustness tests (e.g., against input noise/occlusions) or a small cross-domain experiment.

**Questions:**

1、Key Comparison Details Buried in Appendix. The paper claims a direct comparison with SSCBM under a consistent protocol, but crucial experimental details are relegated to the appendix, making this claim difficult to verify from the main text alone.

2、Central Claim of "Better Uncertainty" is Unproven. The core selling point is that the method produces more reliable uncertainty, yet the paper provides no quantitative evidence. Key metrics like Expected Calibration Error (ECE) or a selective risk analysis are completely missing.

3、Self-Referential Prior. The alignment prior is derived from the same backbone features it is meant to guide. This is a significant methodological flaw, as the prior provides no external information and risks simply amplifying the model's own biases.

4、Localization Claim Lacks Quantitative Support. The assertion that the spatial prior improves concept localization is backed only by qualitative heatmaps. A strong claim like this requires quantitative validation (e.g., pointing accuracy or other localization metrics).

5、Limited Robustness Evaluation. The experiments are confined to standard CBM benchmarks. The paper is missing any analysis of the model's robustness to domain shifts or input perturbations (e.g., noise, occlusions), making its real-world stability unclear.

---

> ### Author Response · Authors · 2025-11-20
>
> We thank the reviewer for the valuable feedback.
>
> Regarding Question 1, we have added key components of the training protocol in the main text (Lines 348--358), while keeping full details in Appendix D. Consistency with SSCBM can now be verified directly from the main paper.
>
> Regarding Question 2, we have added full ECE and selective-risk evaluations in Appendix C (around Line 735). ProCoSA achieves the lowest selective risk across all coverage levels and maintains calibration comparable to the strongest baselines, providing clear quantitative evidence of improved uncertainty reliability.
>
> Regarding Question 3, although the alignment prior uses backbone features, it operates only in the variational E-step and does not affect forward prediction, preventing any self-reinforcing bias loop. It injects local spatial cues complementary to global concept logits and is gated by confidence to act strictly as a stabilizing regularizer. Removing the prior reduces spatial stability, confirming its intended role.
>
> Regarding Question 4, we have added quantitative localization metrics (Pointing Accuracy and IoU) in Appendix G (around Lin 903). ProCoSA achieves the highest pointing accuracy and competitive IoU, demonstrating the effectiveness of the spatial alignment prior beyond qualitative evidence.
>
> Regarding Question 5, we include multiple experiments demonstrating stability under partial concept supervision: theoretical grounding (Line 563), ablations (Tables 4 and 7), calibration and selective-risk results (Table 5), cross-architecture performance (Table 8), and quantitative localization (Table 9). Robustness to domain shift or to input perturbations such as noise and occlusions is an independent research direction that typically requires dedicated datasets and adversarial or stress-test evaluation frameworks. As already stated in the original submission, such robustness analysis is beyond the scope of this work and is highlighted as an explicit direction for future research.
>
> We sincerely thank the reviewer for the time and effort spent on our manuscript and hope that the substantial revisions addressing all concerns are reflected positively in the final assessment.

---

> ### Author Response · Authors · 2025-11-27
>
> Hi, we hope you are doing well. We have updated the submission with the additional experiments and clarifications addressing the raised concerns. Since the discussion period will end in about a week, we just wanted to gently check in to confirm that our responses have been seen. If any further clarification or additional experiments would be helpful, we would be more than happy to provide them. Thank you very much for your time and efforts.

---

### Official Review · Reviewer_5mwT · 2025-11-01

**Soundness:** 2
**Presentation:** 2
**Contribution:** 2
**Rating:** 4
**Confidence:** 2

**Summary:**

This paper proposes ProCoSA, a probabilistic framework for concept-based interpretable learning under partial concept supervision. The motivation is to treat missing concept labels as latent variables and jointly infer concept probabilities and task predictions using an Expectation-Maximization (EM) approach. To improve spatial consistency and reduce pseudo-label bias, ProCoSA introduces a spatial alignment prior that encourages concept embeddings to align with salient image regions. The method is evaluated on 4 datasets (CUB, AwA2, WBCatt, Derm7pt) under low concept supervision ratios (5%–20%), showing consistent improvements in both concept and task accuracy over existing methods like CBM, CEM, and SSCBM.

**Strengths:**

1. ProCoSA treats missing concepts as latent variables and uses variational inference within an EM framework, providing a principled alternative to heuristic pseudo-labeling. Besides, it explicitly models concept uncertainty, which is often overlooked in existing concept bottleneck models (CBMs)

2. ProCoSA provides concept-level saliency maps that align well with human intuition. Besides, it also supports test-time intervention by allowing concept correction, demonstrating causal alignment between concepts and task predictions.

**Weaknesses:**

1. Missing analysis on the computational overhead: the EM loop with multiple fixed-point iterations per E-step may increase training time compared to simpler pseudo-labeling methods, though this is not quantified.

2. While ablations show the impact of alignment and spatial losses, more analysis on the sensitivity to hyperparameters would be useful;

3. ProCoSA lacks a theoretical justification for the convergence of the truncated EM algorithm or the quality of the variational approximations

**Questions:**

Please see the weakness.

---

> ### Author Response · Authors · 2025-11-20
>
> We thank the reviewer for the constructive feedback.
>
> Regarding Question 1, we have added a dedicated analysis in Appendix D(Lines 751--809). The revised manuscript now quantifies ProCoSA's additional training-time cost across datasets and labeled ratios. As clarified there, the overhead comes solely from lightweight E-step updates, which operate on concept logits and alignment scores without requiring extra backbone forward passes. This modest overhead occurs only during training; inference-time cost remains identical to CBM/CEM/SSCBM.
>
> Regarding Question 2, we now include a systematic hyperparameter sensitivity study in Appendix E (Lines 831--858). Varying $\lambda_c$, $\lambda_a$, and $\lambda_s$ across wide ranges shows smooth concept-accuracy changes and extremely stable task accuracy, demonstrating that ProCoSA does not rely on delicate hyperparameter tuning.
>
> Regarding Question 3, we have added a concise theoretical clarification in Appendix A (Lines 563--663). The rule in Eq.~(7) is now explicitly connected to the coordinate-optimal mean-field E-step, and the alignment and consistency terms are explained as regularizers that do not modify the underlying EM decomposition. This provides a clear variational interpretation of the truncated updates.
>
> All revisions have been incorporated into the updated manuscript, and we hope they address the reviewer’s concerns and improve the clarity and completeness of the paper.

---

### Note · Authors · 2025-12-06

I have read and agree with the venue's withdrawal policy on behalf of myself and my co-authors.